# A Pragmatic Ensemble Strategy for Missing Values Imputation in Health Records

**DOI:** 10.3390/e24040533

**Published:** 2022-04-10

**Authors:** Shivani Batra, Rohan Khurana, Mohammad Zubair Khan, Wadii Boulila, Anis Koubaa, Prakash Srivastava

**Affiliations:** 1Department of Computer Science and Engineering, KIET Group of Institutions, Delhi-NCR, Ghaziabad 201206, India; ms.shivani.batra@gmail.com (S.B.); rohankhurana.cse@gmail.com (R.K.); 2Department of Computer Science and Information, Taibah University, Medina 42353, Saudi Arabia; 3Robotics and Internet-of-Things Laboratory, Prince Sultan University, Riyadh 12435, Saudi Arabia; wboulila@psu.edu.sa (W.B.); akoubaa@psu.edu.sa (A.K.); 4Department of Computer Science and Engineering, Graphic Era (Deemed to be University), Dehradun 248002, India; prakash2418@gmail.com

**Keywords:** ensemble learning, health data, imputation methods, missing values, regression algorithms

## Abstract

Pristine and trustworthy data are required for efficient computer modelling for medical decision-making, yet data in medical care is frequently missing. As a result, missing values may occur not just in training data but also in testing data that might contain a single undiagnosed episode or a participant. This study evaluates different imputation and regression procedures identified based on regressor performance and computational expense to fix the issues of missing values in both training and testing datasets. In the context of healthcare, several procedures are introduced for dealing with missing values. However, there is still a discussion concerning which imputation strategies are better in specific cases. This research proposes an ensemble imputation model that is educated to use a combination of simple mean imputation, k-nearest neighbour imputation, and iterative imputation methods, and then leverages them in a manner where the ideal imputation strategy is opted among them based on attribute correlations on missing value features. We introduce a unique Ensemble Strategy for Missing Value to analyse healthcare data with considerable missing values to identify unbiased and accurate prediction statistical modelling. The performance metrics have been generated using the eXtreme gradient boosting regressor, random forest regressor, and support vector regressor. The current study uses real-world healthcare data to conduct experiments and simulations of data with varying feature-wise missing frequencies indicating that the proposed technique surpasses standard missing value imputation approaches as well as the approach of dropping records holding missing values in terms of accuracy.

## 1. Introduction

Amongst the most prevalent problems in data science is the challenge of missing value [1]. This is especially true in health care records, where multiple missing values are common [2,3]. In current history, there is a greater emphasis on ensuring the quality of the data and reusability and automating data discovery and analysis procedures through the publication of data tags and statistical techniques [4]. The creation and use of automated decision support, which can improve reliability, accuracy, and uniformity [5,6], is a fundamental medical application of data science. Substantial training data is often utilised to produce a classifier. In contrast, test data is used to validate system correctness when creating a diagnosing prototype in a clinical decision support system (CDSS) [7]. The training and test data should, in principle, be accurate, with no incomplete data for any parameters. It’s not practicable or viable to get lacking information to enhance data modelling in circumstances of missing value, which frequently happens in real-world traditional therapeutic records. As a result, being the core of their analytical procedure, computational approaches must include a methodology for dealing with missing values.

### 1.1. Motivation

In healthcare prediction, missing data raises serious analytical difficulties. If missing data isn’t treated seriously, it might lead to skewed forecasts. The challenge of dealing with missing values in massive medical databases still needs more effort to be addressed [8]. To minimise the harm to data processing outcomes, it is advisable to integrate multiple known ways of addressing missing data (or design new ones) for each system. The demand for missing data imputation approaches that result in improved imputed values than conventional systems with greater precision and smaller biases is the driving force behind this study.

### 1.2. Missing Data Classification

Little and Rubin [9] described the missing data issue in terms of how missing values are generated, and thus offered three categories: (1) missing completely at random (MCAR), (2) missing at random (MAR), and (3) missing not at random (MNAR). The classification is critical since it influences the prejudices that could exist in the data and the safety of procedures like imputation. When an occurrence lacking a given parameter is unrelated to any other parameter, as well as to the missing values, it is known as missing completely at random (MCAR). It may be claimed that possible occurrences in MCAR are unrelated to any other actual or perceived element in the research. This is the more secure setting in which imputation may take place.

When the likelihood of catching a missing value in a database depends on the observed data of other features and not on the missing data, this is known as missing at random (MAR). MCAR may be thought of as a subset of MAR. Although data MAR has certain ingrained prejudices, it is possible to examine this type of information without specifically correcting for incomplete information. When the chance of the record having a null value is dependent on unseen data, this is known as missing not at random (MNAR). MNAR is prevalent in longitudinal data, such as a medical dataset where illness expansion may result in patients dropping out of the research [10,11]. Longitudinal research studies on mental impairment (i.e., [12,13]) have a high enfeeble rate. In general, medical records are susceptible to the missing value of the MAR variety [14]. However, the likelihood of missing medical data is frequently influenced by the dependent variables since ailment intensity might affect data collection possibilities [15].

### 1.3. Endeavours to Impute Missing Data

A missing value is replaced with appropriate values through data imputation methodologies such as random values, the mean or median, spatial–temporal regressed values, the most common value, or prominent values recognised using k-nearest neighbour [16]. Further, various data imputation methods such as Multivariate Imputation by Chained Equation (MICE) [17] have been established to fill incomplete data numerous times. Deep learning strategies, such as Datawig [18], can predict significantly more precise outcomes than classic data imputation approaches [19] by using the capabilities of GPU and huge data. However, as asserted in the statistical literature [20,21], as the volume of missing data increases, the fluctuation of impact forecasts increases and outcomes may not be accurate enough for hypothesis affirmation if over 40% of values are missing in relevant characteristics [11], implying that data imputation is not a good option when a considerable volume of data is missing. In addition, missing data in the healthcare domain does not happen randomly. Some measured values are missing due to patient discontinuation, medication toxicity, or complicated indicators [22]. Applying MAR data imputation methods to healthcare data may result in biases in forecasting [23].

### 1.4. Importance of Imputing Missing Health Data for Entropy

Entropy is extensively employed in the healthcare field for illness prediction as a nonlinear indicator to quantify the intricacy of the biological system [3]. Aside from the routinely discovered signs, sample entropy can assist doctors in precisely confirming the diagnosis and prediction, allowing them to make better therapy recommendations to patients. However, missing values, which are widespread in the massive volumes of data gathered through medical devices, might make it difficult to use analytic approaches like sample entropy to extract information from them. One research [24] showed that sample entropy can be super vulnerable to missing data and the entropy variations will be substantial once the dataset has missing items. Unfortunately, if the fraction of missing numbers rises, the unexpected variations will rise as well [3]. In order to calculate entropy, it is necessary to handle missing values. Thus, the authors of the current research present a new approach for imputing missing values in health data to reduce the impact of missing data on sample entropy computation.

### 1.5. Research Contributions

Current research provides the following key research contributions.

We introduce a unique Ensemble Strategy for Missing Value to analyse healthcare data with considerable missing values to identify unbiased and accurate prediction statistical modelling. Overall, there are four computational benefits of the suggested model:
It can analyse huge amounts of health data with substantial missing values and impute them more correctly than standalone imputation procedures such as the k-nearest neighbour approach, iterative method, and so on.It can discover essential characteristics in a dataset with many missing values.It tackles the performance glitches of developing a single predictor to impute missing values, such as high variance, feature bias, and lack of precision.Fundamentally, it employs an extreme gradient-boosting method, which includes L1 (Lasso Regression) and L2 (Ridge Regression) regularisation to avoid overfitting.The current study uses real-world healthcare data (snapshot presented in Figure 1) to conduct experiments and simulations of data with varying feature-wise missing frequencies indicating that the proposed technique surpasses standard missing value imputation approaches.

The paper is divided into various sections. Section 2 highlights the related work. Section 3 provides a detailed description of the proposed ensemble method. Section 4 details the experiments conducted, and the results obtained. Section 5 provides a detailed discussion of the current research. Finally, Section 6 concludes this research.

## 2. Related Work

In current history, strategies for dealing with missing values in large datasets have been established. Complete-case analysis (CCA) is the basic and most used technique, which entails deleting the instances containing any missing data and thus concentrating exclusively on individuals who have a complete record for all variables [25]. In fact, because there is typically a large gap between the true distribution of all participants and that of participants with complete details [26], excluding individuals with any missing data would certainly induce biases. Furthermore, the CCA technique will dramatically lower the data size for training prediction models, leading to under-trained frameworks.

Data imputation is another typical approach for dealing with missing data. Single and multiple imputation procedures are the two types of imputation approaches [27]. A single imputation is employed when a missing value can be replaced with an approximated value [28]. The mean imputation [29] replaces a missing item with the mean value. The simple imputation technique has the drawback of drastically underestimating data variation and ignoring intricate interactions among potential determinants [25].

For missing value imputation, the k nearest neighbours (kNN) approach is often employed. The kNN imputation method substitutes mean values from k closest neighbours for relevant attributes. Many studies have been conducted to increase kNN’s imputation accuracy. To improve imputation efficiency, Song et al. took comparable neighbourhoods into consideration [30]. More advanced single imputation strategies, including regression imputation and expectation–maximization (EM), can be used to resolve this issue [29]. Regression models were used as a substitute to repair missing values in [31]. Instead of attempting to deduce missing values, Song et al. [32] recommend first estimating lengths between absent and entire values, and then imputing values using inferred lengths. These techniques allocate a missing value by analysing the correlations between the dependent attribute and the remaining parameters in the dataset. Chu et al. [33] focused on data cleaning approaches, including various functional dependencies in a unified framework. Breve et al. [34] proposed a novel data imputation technique, based on relaxed functional dependencies, that identifies value possibilities that effectively ensure data integrity. However, in the case of healthcare data, we often encounter temporal functional dependencies for the data of patients collected for a time span [35].

On the other hand, numerous imputation approaches use multiple imputed values to approximate a missing value. Multivariate imputation by chained equations (MICE) is an approach in which the statistical uncertainties of diverse imputed data are properly considered [36]. Unfortunately, for every database, neither of the available imputation methods beats all others, implying no standard framework [29] for missing value imputation.

Although most machine learning techniques can only be used to impute missing data or to employ CCA by default [29], XGBoost [37,38], a modern version of the gradient-boosting technique, has crafted features that can autonomously manage missing data. XGBoost addresses the issue of missing values by including a pre-set path for missing data in each tree split. During the training phase, the best path for a missing value in every explanatory parameter at every node is discovered with the objective of minimizing the regulatory losses [37]. If no missing data in any explanatory parameter exists in the training examples, but there are missing values in the testing dataset, the XGBoost model takes the pre-set path. The pre-set path for parameter estimates on the testing set is often chosen by XGBoost, which might be a concern when dealing with missing values in XGBoost. If the missing data trends in the training and test dataset are dissimilar, the forecast might be a rough estimate. This might be the scenario if there is a significant quantity of missing value, particularly in the test dataset.

Overall, conventional machine learning algorithms face the challenge of not being adaptable enough to handle big missing data. Furthermore, the disparity across training and test data has not been adequately addressed when it comes to model inference. An ensemble model for data imputation is introduced in this paper. Ensemble models are a machine learning methodology that combines numerous different models to provide a forecast. Other models involved in the ensemble model are referred to as base predictors. Ensemble approaches benefit from boosting poor learners to turn into leading ones [39,40]. Ensemble approaches have been utilized in a variety of domains to improve the accuracy of the system. Troussas et al. [41] suggested an ensemble classification approach that uses the support vector machine, naive bayes, and KNN classifiers in combination with a majority voting mechanism to categorise learners into appropriate learning styles. The model is first trained using a collection of data, and then the category of the occurrence is forecasted using the base classifiers with the majority of votes. Zaho et al. [42] devised an ensemble technique by integrating patch learning with dynamic selection ensemble classification, wherein the miscategorised data have been used to educate patch models in order to increase the variety of base classifiers. Rahimi et al. [43] used ensemble deep learning approaches to construct a classification model that improved the accuracy and reliability of classifying software requirement specifications.

Authors have devised a pragmatic ensemble technique for missing value imputation based on the same concept. The below-listed technological obstacles of developing a single imputer are likewise solved by this strategy.

High variance is achieved by rendering the model supersensitive to the inputs given to the acquired characteristics.Inaccuracy due to fitting the intensive training data with a single model or technique may not be sufficient to satisfy expectations.When making predictions, noise and bias cause the models to rely mainly on one or a few features.

## 3. Materials and Methods

Ensemble learning is an amalgamation of various machine learning techniques that contemplates the estimate of various base machine learning models (base estimators) in order to achieve better predictive performance. As a base estimator, one can implement any machine learning algorithm. If the nature of considered base learners is homogeneous then the ensemble strategy is termed a homogeneous ensemble learning method, otherwise, the ensemble strategy is termed non-homogeneous or heterogeneous. The ensemble machine learning can be constructed on three sorts of mechanisms viz. bootstrap aggregation (Bagging), boosting, and stacking. Bootstrap aggregation comprises independently learning weak learners (base estimators) and the outcome is the average of resultants calculated by different weak learning. While in boosting mechanism, the base estimators are summarized one after the other and then resultant is generated as the weighted average of base estimators’ outcomes. On the other hand, stacking ensemble mechanism fed the same data to all chosen base estimators and then trains an additional machine learning model called a meta-learner to upgrade model’s overall performance. In this research, the authors have employed the stacking mechanism of ensemble strategy in order to devise a novel methodology of missing data imputation for Health Informatics. This research will be using different stand-alone imputations as individual base estimators in Level 1 and then combining the outcomes of these base estimators and feeding them to a meta learner machine learning model in Level 2 to make the final predictions. Figure 2 illustrates the conceptual schema of the proposed ensemble strategy.

The proposed ensemble approach targets to discover unbiased and accurate prediction trends from healthcare data, which, if trained directly, might lead to biases due to significant missing values [44]. The suggested model has three stages:Data pre-processingModel trainingImputation

Hereafter in this research, the authors will be using the $D\in {ℛ}^{M\times N}$ matrix to represent the dataset, which has M observations and N characteristics. Further, $d_{i,j}$, which is the parameter value for the *j*th characteristic of *i*th observation, is the item of D at position (*i*, *j*). Many parameters’ values are missing because of several intercurrent occurrences, including medication suspension or early cessation for multiple causes. The features that hold missing values have been discovered, and their feature indices have been placed in vector Q. In addition, $\bar{Q}$ is a vector that represents feature indices which do not include any missing values. Also, $D^{train}$ dataset consists of *p* samples with no missing values in any of the rows.

### 3.1. Data Pre-Processing

In the data pre-processing phase, the raw data is processed to produce training data that will be used as input to a regressor model. Figure 3 depicts the entire data pre-processing procedure, which is accomplished as listed below.

Initially, the training data i.e., $D^{train}$, does not contain any missing values. Thus, a dataset, i.e., $D^{train_{mv}}$, is prepared by randomly eliminating the present values from the features present in Q.Three imputation techniques were chosen for the proposed ensemble methodology as unrelated base predictors since using unrelated base predictors may significantly reduce prediction errors in ensemble learning, as indicated in [40]. $D^{train_{mv}}$data is passed to three imputation methods, i.e., (1) simple mean imputer, (2) KNN imputer, and (3) iterative imputer, that have been chosen as base predictors in current research.
**Simple mean imputer:** Missing values are substituted in this imputer by the mean of all non-missing values in the corresponding parameter.**KNN imputer:** By assessing respective distance measurements, the KNN method seeks the other *k* non-missing findings, most comparable to the missing one for every missing value. The missing data is subsequently replaced by a weighted average of the *k* nearby but non-missing values, with the scores determined by their Euclidean distances from the missing value.**Iterative imputer:** Multiple copies of the same data are generated and then integrated to get the “finest” predicted value in this approach. The MICE technique has been used to provide iterative imputation based on completely conditional requirements.The values predicted to be imputed for the missing data in $D^{train_{mv}}$ by the base predictors are reserved in three 2-D matrices, i.e., $Pred^{1},\;Pred^{2},\;and\;Pred^{3},$ for simple mean, KNN, and iterative imputer, respectively.Corresponding to each attribute index in Q, a regressor model is trained. For training each q ϵ{1,2,…,Q}  regressor models, a corresponding matrix $P_{q}$ (structure presented in Equation (1)) is provided as input.

(1)$$P_{q}=\left\{Pred_{p,q}^{1},\;Pred_{p,q}^{2},Pred_{p,q}^{3},D_{p,q}^{train}\right\},\;p\;\in all\;samples$$
where, $Pred_{p,q}^{1},\;Pred_{p,q}^{2},\;\mathrm{and}\;Pred_{p,q}^{3}$ represents the value of *q*th attribute of *p*th sample imputed by simple mean, KNN, and iterative imputer, respectively, and $D_{p,q}^{train}$ depicts the actual known value of *q*th attribute of *p*th sample.

### 3.2. Model Training

The proposed ensemble model employs the eXtreme Gradient Boost (XGB) regression technique for training purposes. An XGB Model is trained for each attribute index in Q. Thus, there will |Q| XGB models. As detailed in the previous section, the training data $P_{q}\;$(for q ϵ{1,2,…,Q}) is provided as input to each XGB regression model for model training, as depicted in Figure 4.

The value of *i*th entry of $P_{q}$, i.e., $\;{\hat{a}}_{i}$, is predicted using Equation (2), where $a_{i}$ is the observed value of *i*th entry and $b_{i}$ is the sample input corresponding to {$P_{q}^{i,1},\;P_{q}^{i,2},P_{q}^{i,3}$}.
(2)$${\hat{a}}_{i}=\sum_{t=1}^{T}score_{t}\left(b_{i}\right),\;where\;score_{t}\in T$$

The function $score_{t}$ presents an independent tree among the set of regression trees, T and $score_{t}\left(b_{i}\right)$ refers to the anticipated score provided by the *i*th sample and *t*th tree. The objective function of the XGB, designated by $O_{XGB}$, is calculated as presented in Equation (3):(3)$$O_{XGB}=\sum_{i=1}^{n}\Delta \left(a_{i},{\hat{a}}_{i}\right)+\sum_{t=1}^{T}\chi \left(score_{t}\right)$$

The regression tree model functions $score_{t}$ can be trained by minimizing the objective function, $O_{XGB}$. The gap between the forecasted value, ${\hat{a}}_{i}$ and the true value, $a_{i}$ is evaluated by the training loss function $\Delta \left(a_{i},{\hat{a}}_{i}\right)$. Further, χ is employed to prevent the challenge of overfitting by penalising model intricacy as presented in Equation (4) for the independent tree, *t* among the set of regression trees.
(4)$$\chi \left(score_{t}\right)=\varphi \xi +0.5*\;\eta \Theta^{2}$$
where φ and η are the regularization factors. φ dictates if a particular node split depending on the anticipated loss minimization after the split, and η is L2 regularisation on leaf weights. ξ and Θ are the numbers of leaves and scores on every leaf, respectively. The objective function can be approximated using a second-degree Taylor series [45]. Further, summation is a useful mechanism to train the ensemble model. Let $\Phi_{j}=\text{\{}i\;|t\left(b_{i}\right)=j\}$ be an occurrence set of leaf *j* with the fixed structure t(b). The Equation (5) is used to calculate the optimum weights $\Theta_{j}{}^{*}$ of leaf *j* using first and second gradient orders of loss function and also the optimum value of associated loss function $O_{XGB}{}^{*}$.
(5)$$\Theta_{j}{}^{*}=-\frac{r_{j}}{s_{j}+\eta },\;and{\;O}_{XGB}{}^{*}=-0.5*\sum_{j=1}^{\xi }\frac{{\left(\sum_{i\in \Phi_{j}}r_{j}\right)}^{2}}{\sum_{i\in \Phi_{j}}s_{i}+\eta }+\eta \xi$$

The first and second gradient orders of the loss function, $O_{XGB}$ are $r_{i}$ and $s_{i}$, respectively. Further, $O_{XGB}$ can be used to discover the quality score for *t*. The lower the score, the more accurate the model. Because computing all the tree topologies is impossible, a greedy approach is employed to tackle the issue, starting with only one leaf and repeatedly extending paths to the tree.

After splitting, let $\Phi_{L}$ and $\Phi_{R}\;$be the occurrence sets of the left and right nodes, respectively. If the original set is Φ = 

$\Phi_{L}\cup \Phi_{R}$, the loss reduction following the split, $O_{XGB\_split}$ will be as presented in Equation (6).
(6)$$O_{XGB\_split}=0.5*\left[\left\{\sum_{i\in \Phi_{L},{\;\Phi }_{R}}\left(\frac{{\left(\sum_{i}r_{i}\right)}^{2}}{\sum_{i}s_{i}+\eta }\right)\right\}-\frac{{\left(\sum_{i\in \Phi }r_{i}\right)}^{2}}{\sum_{i\in \Phi }s_{i}+\eta }\right]-\varphi$$
where the first term depicts the summation of score associated with left and right leaf, second term depicts the score associated with the original leaf, i.e., leaf before the splitting operation is performed and φ is the regularisation term on additional leaf that will be used further in the training process. In practice, this approach is commonly used to evaluate split candidates. During splitting, the XGB model employs many simple trees, as well as the leaf node similarity score.

### 3.3. Imputation

Utilising the trained ensemble model, the missing values are imputed for the test dataset. The test data is represented as $D^{M\times N}$, which has *M* instances and *N* attributes, with Q being the attribute with at least one missing value. The dataset is pre-processed in the same fashion as in the first phase of the proposed model, with the exception that there will only be three columns since the actual value is to be anticipated, resulting in a two-dimensional matrix of the form presented in Equation (7).
(7)$$P_{q}{}^{test}=\left\{Pred_{m,q}^{1^{test}},\;Pred_{m,q}^{2^{test}},Pred_{m,q}^{3^{test}}\right\},\;m=\left\{1,2,\ldots ,\;M\right\},\;q\in Q$$
where, $Pred_{m,q}^{1^{test}},\;Pred_{m,q}^{2^{test}},\;and\;Pred_{m,q}^{3^{test}}$ are the value of *q*th attribute of mth sample, imputed by the base predictors, i.e., simple mean, KNN, and iterative imputer, respectively.

The missing value within every feature may be simply inferred using equation (2) with the support of trained ensemble models. Let $Y_{q\;}$ denote the vector holding the anticipated values of the proposed ensemble model’s *q*th XGB regressor, as shown in Figure 5.

Using the anticipated set of vectors, $Y_{1\;},Y_{2\;},\ldots ,\;Y_{q}$ (as presented in Equation (8)), the missing values are imputed in test dataset D.
(8)$$D\left[m\right]\left[q\right]=\{\begin{array}{c} Y_{q}\left[m\right],\;if\;D\left[m\right]\left[q\right]=nan\\ D\left[m\right]\left[q\right],\;otherwise \end{array}\;$$
where m={1,2,…,M}, q={1,2,…, Q}, nan=missing value or empty. Algorithm 1 presents procedure of proposed ensemble model. The algorithm has been partitioned into three sections, i.e., variable declaration, generation of training dataset, then training the model and applying trained model to the testing dataset.

In the first section (variable declaration), all the required datasets and matrices have been initialised.In the second section, the algorithm performs two sequential tasks.
The first task involves generation of training dataset using three imputation strategies, i.e., simple imputation, kNN imputation, and iterative imputation; after applying imputation method on the training dataset, the resultant dataset is stored in Pred1, Pred2, and Pred3, respectively. Now, for each attribute index present in Q, a corresponding matrix Pq is formed that comprises of four attributes (simple, kNN, iterative, and actual). The first three attribute elements are represented by vector B denoting the values of *q*th attribute’s elements imputed by simple imputation, kNN imputation, and iterative imputation method, and the fourth attribute element is represented by vector A, denoting the known value of *q*th attribute’s elements.The second task involves the training of a regressor model (XGB) using generated training dataset. The vectors B and A are passed into XGBRegressor method for training the model and the trained resultant regressor associated with the *q*th attribute is represented by reg[*q*].In the third section, the algorithm performs three sequential tasks.
The first task involves the preprocessing of the testing dataset as done in previous section and transform testing dataset representation into Pqtest matrix associated with each missing valued attribute (*q*). Pqtest matrix comprises of three attribute elements represented by vector Btest denoting the values of *q*th attribute’s elements imputed by simple imputation, kNN imputation, and iterative imputation methods.
**Algorithm 1** Proposed Ensemble ModelD: testing dataset, Q: dataset with imputed instancesQ: indexes of attributes having at least one MV.Dtrain: dataset with training instances.Dtrainmv: dataset with training instances having randomly assigned MVs.reg[*q*]: regressor model associated with *q*th feature*#Generating training Dataset and training the Model*Pred1= SimpleImputer(Dtrainmv, strategy = ‘mean’)Pred2= kNNImputer(Dtrainmv, NN = 5)Pred3=IterativeImputer(Dtrainmv, max_itr = 5)for *q*th in Q: Pq[0]=Pred1[q], Pq[1]=Pred2[q], Pq[2]=Pred1[q], Pq[3]=Dtrain[q] B=(Pq[0],Pq[1],Pq[2]) A=(Pq[3])reg[*q*] = XGBRegressor()reg[*q*].fit(B,A)reg[*q*].predict(B)*#Applying trained ensemble models on*
DPred1test= SimpleImputer(Dtrainmv, strategy = ‘mean’)Pred2test= kNNImputer(Dtrainmv, NN = 5)Pred3test= IterativeImputer(Dtrainmv, max_itr = 5)for *q*th in Q: Pqtest[0]=Pred1test[q], Pqtest[1]=Pred2test[q], Pqtest[2]=Pred3test[q] Btest=(Pqtest[0],Pqtest[1],Pqtest[2]) Btest = Btest[D[*q*].isna().index] Yq = reg[*q*].predict(Btest) i=−1 for *j* in D[*q*]:  if D[*q*][*j*] = nan:   D[*q*][*j*]= [*i*++]
b.The second task involves the prediction of missing values in testing dataset using trained regressor models (XGB) reg[*q*] associated with each missing valued attribute (*q*). The predicted values are stored in a vector Yqc.Lastly, the third task involves the substitution of imputed results of missing values associated with *q*th attribute as stored in Yq into the actual dataset D. After substitution, the dataset is completed, and no missing value is present in it.

## 4. Experiments and Results

The experimental environment was a PC with an Intel(R) Core^(TM)^ i3-6006U CPU @ 2.00 GHz running the Windows 10 operating system with 11.9 GB RAM. This research utilised XGB, Support Vector, and Random Forest Regressor to quantify the accuracy of the decision support system provided after imputing the missing values through the underlying imputation approach to assess the proposed ensemble imputation technique with a simple mean, kNN, and multiple imputation methodologies. Table 1 lists the configurations of the three regressors and four imputation techniques. Further, the experiments are also conducted on the dataset by simply dropping the missing value to assess its effects on prediction in comparison to the proposed ensemble method.

### 4.1. Real Time Dataset

This research utilised the real-time COVID-19 epidemic dataset [46], which included missing values with varying missing percentages, and varying quantities of characteristics and occurrences for the experimentation. This real-time dataset contains information on the COVID-19 epidemic in the United States, with records from 3142 US jurisdictions from the start of the epidemic (January 2020) through June 2021. This information refers to different publicly accessible databases and encompasses the everyday count of COVID-19 confirmed incidence and mortality, as well as 46 other attributes that could affect pandemic trends, such as each county’s demographic, spatial, environmental, traffic, public health, socioeconomic compliance, and political characteristics. The underlying dataset constitutes 750,938 records and 58 attributes, among which 12 attributes hold missing values. A total of 10K records are chosen randomly from the original dataset for model training. Further, three varying sizes (5K, 10K, and 20K records) of the dataset are chosen randomly for testing the proposed model. The randomly chosen test data is statistically analysed to quantify the missing values present, as depicted in Table 2. Moreover, missing values are observed in each attribute (i.e., 12 attributes holding missing values) for every varying size test dataset as presented in Table 3 and Figure 6.

### 4.2. Regressor Models

For determining the performance of the proposed ensemble framework, the authors have selected three regression models, i.e., Support Vector Regressor (SVR), Random Forest Regressor (RFR), and eXtreme Gradient Boost Regressor (XGBR). These regression models are built to check the performance of different missing data-handling methodologies discussed in the paper (i.e., proposed ensemble imputation method, simple mean imputation method, kNN imputation method, and iterative imputation method). The covid_19_deaths attribute is chosen as the target attribute to train and test these models because it has no missing values and it also happens to be the target variable in the dataset [46]. The regressor models are briefly illustrated as follows:**eXtreme Gradient Boost Regressor (XGBR):** XGBoost is a tree-based enactment of gradient boosting machines (GBM) utilised for supervised machine learning. XGBoost is a widely used machine learning algorithm in Kaggle Competitions [47] and is favoured by data scientists as its high execution speed beats principal computations [37]. The key concept behind boosting regression strategy is the consecutive construction of subsequent trees from a rooted tree such that each successive tree diminishes the errors of the tree previous to it so that the newly formed subsequent trees will update the preceding residuals to decrease the cost function error. In this research, the XGB Regressor Model has a maximum tree depth of 10, and L1 and L2 regularisation terms on weights are set as default, i.e., 0 and 1, respectively.**Random Forest Regressor (RFR):** Random Forest is an ensemble tree-based regression methodology proposed by Leo Breiman. It is a substantial alteration of bootstrap aggregating that builds a huge assemblage of contrasting trees, and after that aggregates them [48]. A random forest predictor comprises an assemblage of unpremeditated regression trees as the base $\left\{T_{i}\left(A,\Psi_{j},D_{i}\right)\right\}$, where $\Psi_{1},\Psi_{2},\ldots ,\;\Psi_{j}$, are independent and identically distributed (IID) outcomes of a randomising variable Ψ and j≥1. An aggregated regression estimate is evaluated by combining all these random trees by using formula $\bar{T_{i}}\left(A,D_{i}\right)=E_{\Psi }\left[T_{i}\left(A,\Psi_{j},D_{i}\right)\right]$, where $E_{\Psi }$ denotes expectation w.r.t. with the random variable conditionally on A and the dataset $D_{i}$. In this research, the maximum depth of RFR tree is tuned to 5, and other parameters, such as the minimum sample split and the number of trees, are kept as the default, i.e., 2 and 1000, respectively.**Support Vector Regressor (SVR):** Support Vector Machine (SVM) used for regression analysis is named as support vector regressor (SVR) [49]. In SVR, the input values are mapped into a higher-dimensional space by some non-linear functions called kernel functions [50,51] so as to make the model linearly separable for making predictions. The SVR model is trained by a structural risk minimisation (SRM) principle [52] to perform regression. This minimises the VC dimension [53] as a replacement for minimising the mean absolute value of error or the squared error. In this research, SVR uses the radial basis function as kernel and a regularisation parameter (C) of 1.5.

### 4.3. Evaluation Metrics

An evaluation is a common method of determining a model’s performance. After imputation of the missing values, this research employed eXtreme Gradient Boost, Support Vector Machine, and Random Forest regressors to determine the desired values, with mean absolute error (as depicted in Equation (9)) and mean squared error (as depicted in Equation (10)) used to assess correctness where $a^{\mathrm{imputed}}$ and $a^{\mathrm{actual}}\;$are the imputed and actual value for *p* records.
(9)$$\mathrm{mean}\;\mathrm{absolute}\;\mathrm{error}=\frac{\sum_{i=1}^{p}\left|a_{i}{}^{\mathrm{imputed}}-a_{i}{}^{\mathrm{actual}}\right|}{p}$$
(10)$$\mathrm{mean}\;\mathrm{squared}\;\mathrm{error}=\frac{\sum_{i=1}^{p}{\left(a_{i}{}^{\mathrm{imputed}}-a_{i}{}^{\mathrm{actual}}\right)}^{2}}{p}$$

### 4.4. Results

As stated above, experiments are conducted using three regressors, i.e., XGB Regressor, Support Vector Regressor (SVR), and Random Forest Regressor (RFR), for varying sizes of test data (i.e., 5000, 10000, and 20,000 records) employing four imputation methods (i.e., proposed ensemble, iterative, kNN, and simple mean) and simply dropping the instances holding missing values. The results obtained are presented in Table 4 and Figure 7 in terms of two evaluation metrics, i.e., mean absolute error and mean squared error.

To generalize the evaluation metrics for comparison, in each regression model authors have normalised the resultant value of all underlying imputers with respect to the resultant value of the proposed ensemble model as devised in Equations (11) and (12).
(11)$${\left({\mathrm{mean}\;\mathrm{absolute}\;\mathrm{error}\;}_{normalized}\right)}_{regressor}=\frac{{\left(\mathrm{mean}\;\mathrm{absolute}\;\mathrm{error}\;\right)}_{imputationMethod}}{{\left(\mathrm{mean}\;\mathrm{absolute}\;\mathrm{error}\;\right)}_{proposedMethod}}$$
(12)$${\left({\mathrm{mean}\;\mathrm{squared}\;\mathrm{error}}_{normalized}\right)}_{regressor}=\frac{{\left(\mathrm{mean}\;\mathrm{squared}\;\mathrm{error}\right)}_{imputationMethod}}{{\left(\mathrm{mean}\;\mathrm{squared}\;\mathrm{error}\right)}_{proposedMethod}}$$
where, imputation method ϵ {Iterative, KNN, Simple Mean, Dropping Instances} and regressor ϵ {XGB, RFR, SVR}. If the normalised value is obtained as 1, the performance of the underlying imputation technique is identical to the proposed ensemble model. Further, if the normalised value is greater than 1, the corresponding imputation approach outperforms the proposed ensemble model; otherwise, the underlying imputation technique underperforms in comparison to the proposed ensemble model. The observed normalised values are presented in Table 5.

There are two key conclusions based on the experimental comparison for the proposed ensemble model presented in Table 5 and the graphical analysis illustrated in Figure 8.

It has been discovered that primitive imputation strategies, such as iterative, kNN, and simple mean imputation do not perform well when imputing the missing values of huge datasets. When the imputed dataset is submitted to XGB regressor and random forest regressor to assess target values, dropping the records with missing values appears to be highly promising, as demonstrated in Table 5. On the contrary, while making predictions through a support vector regressor, dealing with a large dataset containing comparatively more missing values, dropping the missing values is not recommended. However, when the dataset is small and has fewer missing values, dropping the records holding missing values is the best option, as predicted by all three regression models.When working with a small dataset with fewer missing values, all imputation techniques produce similar outcomes when predicted by the SVR Model. On the contrary, in the case of regressor models XGB and RFR, significant variations in the performance of various imputation techniques are observed. The results achieved indicate that the proposed ensemble model outperforms all mentioned primitive imputation techniques when dealing with both large and small datasets by producing the lowest values for mean absolute and mean squared errors. The performance of kNN, iterative, and simple mean imputation to impute missing values individually has been observed to underperform compared to the technique of dropping the records holding missing values. However, the suggested ensemble imputations model outperformed all four scenarios, as validated by the three underlying regression models.

## 5. Discussion

After analysing the evaluation metrics generated by three regressor models, it has been found that the proposed ensemble strategy is the most suitable option for the imputation of missing values. The imputed dataset produced by the Proposed Ensemble approach when passed to XGB Regressor for performance evaluation results in the least mean absolute error, i.e., 60.81, 54.06, and 49.38, and least mean squared error, i.e., 8266.08, 6046.26, and 4473.7, in all three test cases considered. Similarly, when the same dataset is passed to the RFR model, the model gives the least mean absolute error, i.e., 112.8, 115.98, and 113.57, and the least mean squared error, i.e., 23,966, 23,256.3, and 23,298.47, in all three test cases considered. However, when the same imputed dataset is passed to the SVR Model, then in one of the test cases, i.e., with 20,000 records, it gives the least mean absolute error of 188.31, and in two cases, i.e., with 10,000 and 20,000 records, it gives least means squared error of 63,853.1 and 59,422.4, respectively, as represented in Figure 8.

For the comparison of state-of-the-art missing value-handling strategies such as simple imputation, kNN imputation, iterative imputation, and dropping the missing value contained instances method, normalised error results have been calculated using Equations (11) and (12) with respect to the proposed imputation method as depicted in Table 5. It has been observed that the approach of dropping the instances with missing values is the closest missing value handling method to the proposed ensemble model as it results in the normalised error estimate in the range of 0.7 and 1.0 in all three considered test cases. But the method reduces the dataset size, thus it should not be preferred for large and crucial datasets.

On the other hand, among simple mean, kNN, and iterative methods, iterative imputation is closest to the proposed imputation method having a normalised MAE of 0.775, 0.742, and 0.679 in the three considered test cases, i.e., 5000, 10,000, and 20,000 records, respectively, and a normalised MSE of 0.593 and 0.473 in two test cases, i.e., 10,000 and 20,000 records, respectively, as computed by XGB Regressor Model. On the contrary, the simple mean imputation method is closest to the proposed imputation method having a normalised MAE 1.023, 1.011, and 0.994 and a normalized MSE 1.021, 0.981, and 0.954 in the three considered test cases, i.e., 5000, 10,000, and 20,000 records, respectively, as predicted by the SVR Model and a normalised MAE and normalised MSE of 0.768 and 0.678 as predicted by RFR and XGB Model. Similarly, the kNN imputation method closest to the proposed imputation method having a normalised MAE 0.782 in one test case, i.e., 20000 records, and a normalised MSE of 0.634 and 0.627 in the two considered test cases, i.e., 5000 and 20,000 records, respectively, as predicted by RFR Model. Hence it can be said, when the dataset size is small and has fewer missing values, dropping the records holding the missing values seems the most suitable approach, as predicted by almost all three regression models, and with a large dataset size the simple mean, kNN, and iterative method give equivalent results in most of the cases but could not match with the performance of the proposed ensemble strategy as estimated by considered regressor models.

In current research, authors are focused on establishing an ensemble technique for missing value imputation employing mean value, kNN, and iterative imputation techniques. However, in the near future, authors aim to extend the current research on the below-listed limiting parameters of the proposed model.

**Functionally dependent domain:** Current research is not exploiting the functional dependencies present in the dataset for identification of missing values. The authors target to employ the devised ensemble strategy on other healthcare datasets including genomics-based and specific disease diagnosis-based, which may include the significance of attribute’s functional dependencies.**Intelligent selection of base predictors**: The base predictors chosen in the proposed model are fixed and thus do not consider other base predictors available. The authors intend to develop a system for intelligent selection and hybridisation of the different base estimators on the basis of attributes, for instance, domain dependency; categorical data must be addressed by classification-based machine learning models and continuous data must be addressed by regression machine learning models. Further, the multiple stacking approach can be integrated for the meta learners in the proposed ensemble approach, wherein the XGB model can be replaced with the kNN-based deep learning methods when handling complex healthcare datasets which can help in producing much better outcomes and can be more reliable in terms of performance.

## 6. Conclusions

To efficiently model computer systems to aid in medical decision-making, clean and reliable data is essential, yet data in medical records is usually missing. Leaving a considerable amount of missing data unaddressed frequently results in severe bias, which leads to incorrect conclusions being reached. In the current research work, an ensemble learning framework is introduced that (1) can handle large numbers of missing values in medical data, (2) can deal with various datasets and predictive analytics, and (3) considers multiple imputer values as base predictors, utilising them to construct new base learners for the entire ensemble that result in maximum correlation value with respect to the negative gradient of the loss function. The performance of the proposed ensemble method has been evaluated compared to three commonly used data imputation approaches (i.e., simple mean imputation, k-nearest neighbour imputation, and iterative imputation) and a basic strategy of dropping records containing missing values in the experiments conducted. Simulations on real-world healthcare data with varying feature-wise missing frequencies, number of instances, and three different regressors (eXtreme gradient boosting regressor, random forest regressor, and support vector regressor) revealed that the proposed technique outperforms standard missing value imputation approaches.

## Figures and Tables

**Figure 1 entropy-24-00533-f001:**
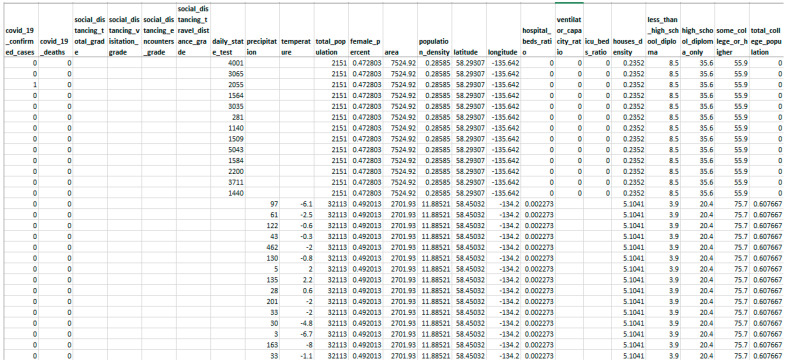
Snapshot of sample real-world data explored for experimentation.

**Figure 2 entropy-24-00533-f002:**
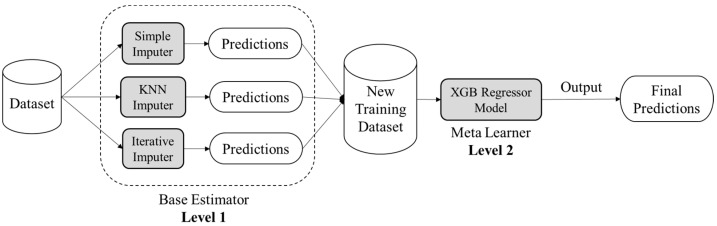
Conceptual schema of proposed ensemble approach based on stacking mechanism.

**Figure 3 entropy-24-00533-f003:**
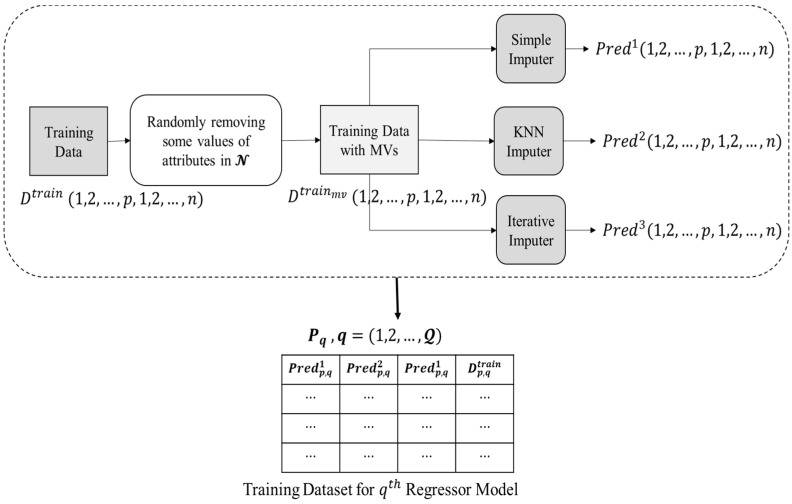
Data pre-processing phase.

**Figure 4 entropy-24-00533-f004:**
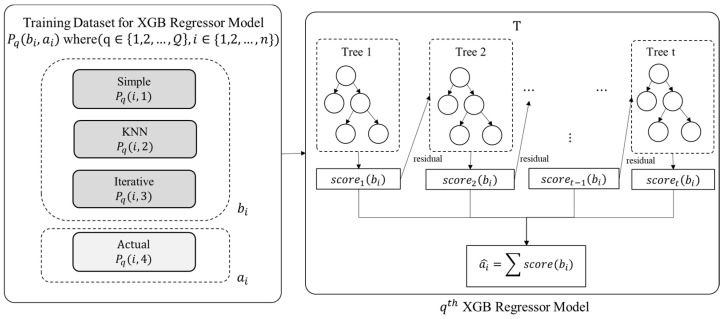
Model Training Phase.

**Figure 5 entropy-24-00533-f005:**
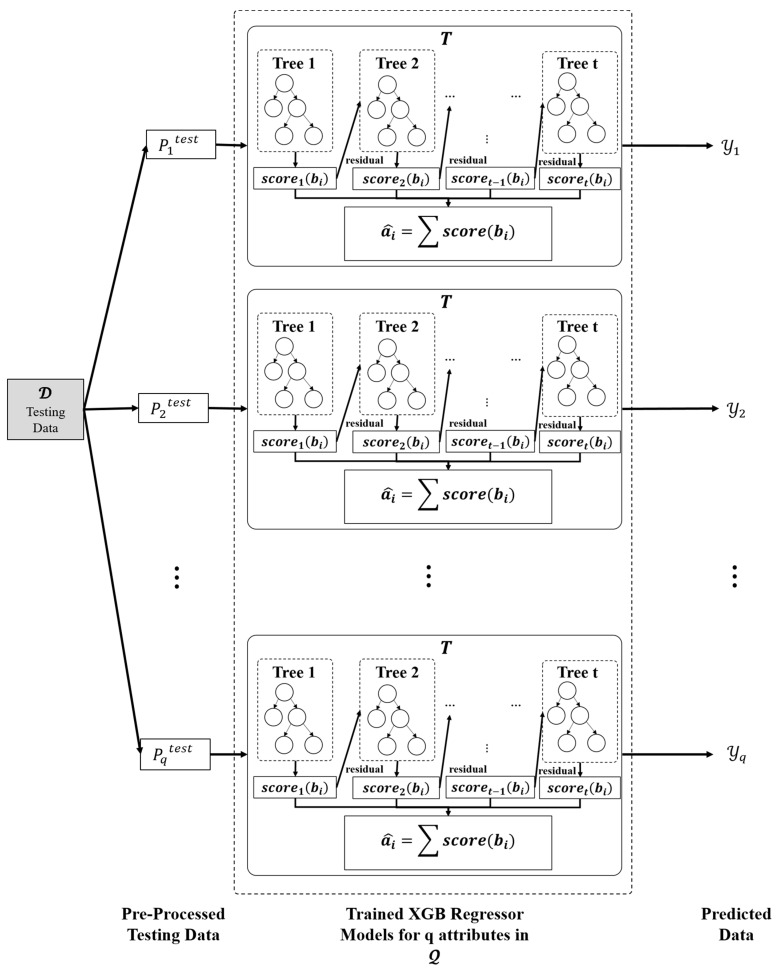
Imputation phase.

**Figure 6 entropy-24-00533-f006:**
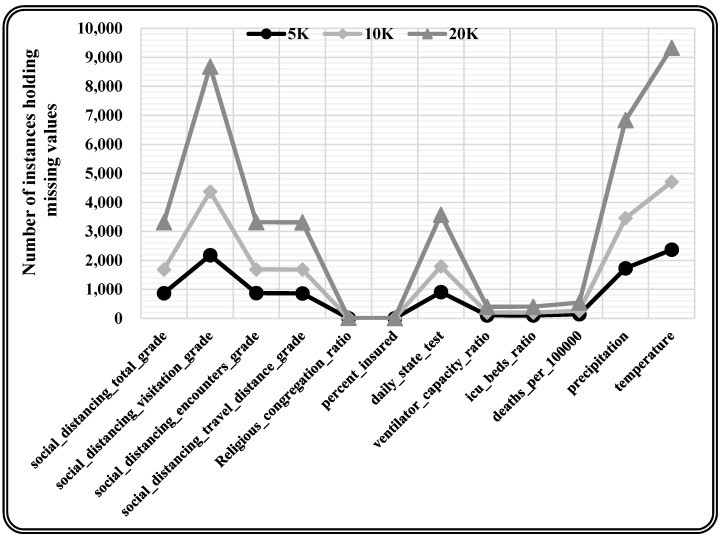
Graphically presented attribute-wise missing values for varying size test dataset.

**Figure 7 entropy-24-00533-f007:**
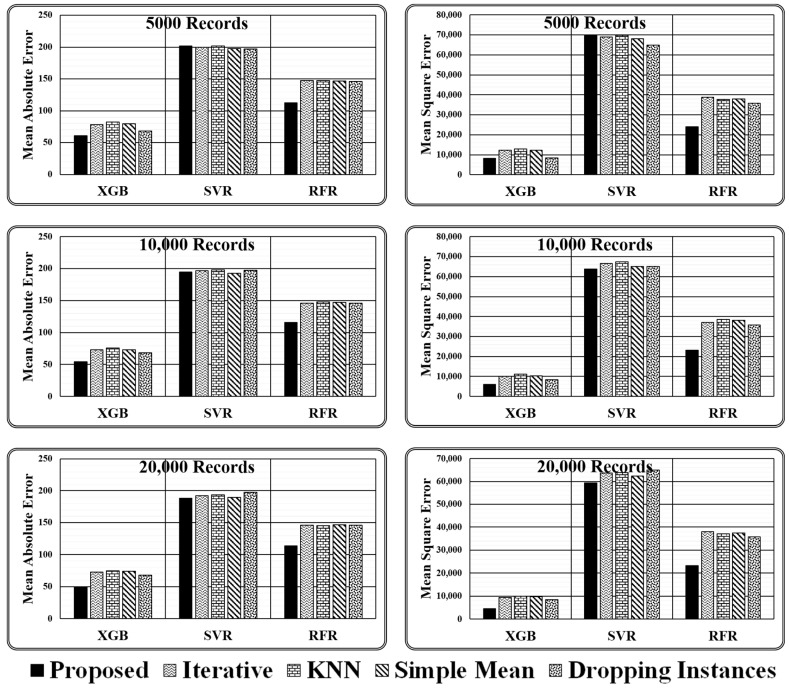
Graphical presented results obtained for varying size test dataset.

**Figure 8 entropy-24-00533-f008:**
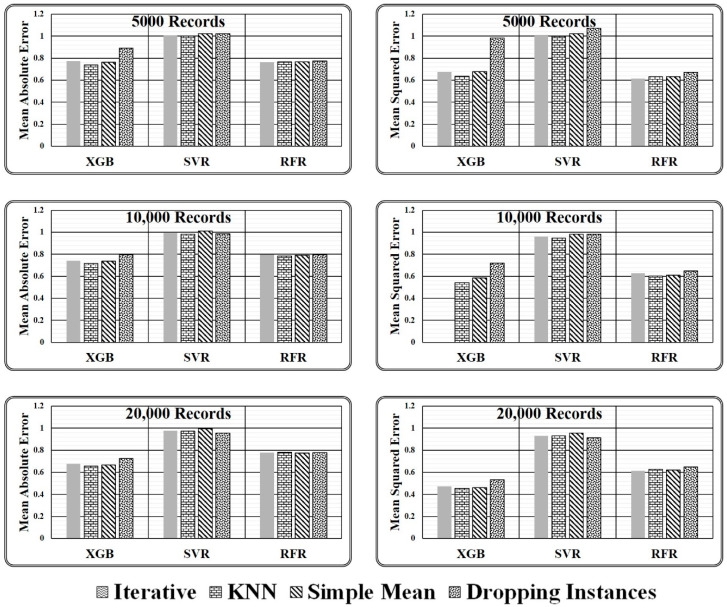
Graphical presented normalised results obtained for varying size test dataset.

**Table 1 entropy-24-00533-t001:** Configurations of regressors and imputation techniques.

Regressor/Imputation Methods	Configurations
XGB Regressor	max_depth = 10
Support Vector Regressor	Kernel = rbf, C = 1.5
Random Forest	max_depth = 5
K Nearest Neighbour Imputation	K = 5
Multiple Imputation	max_itr = 5
Simple Imputation	strategy = ‘mean’
Proposed Ensemble Model Imputation	NA

**Table 2 entropy-24-00533-t002:** Instances holding one or more missing values in test dataset.

Test Dataset Size	Number of Instances Holding One or More Missing Values	Frequency of Non-Missing Values	Frequency of Missing Values
5000	3458	279,877	10,123
10,000	6961	559,955	20,045
20,000	13,857	1,120,278	39,722

**Table 3 entropy-24-00533-t003:** Attribute-wise missing values for varying size test datasets.

Attributes Name	5K Records	10K Records	20K Records
social_distancing_total_grade	868	1682	3315
social_distancing_visitation_grade	2176	4369	8681
social_distancing_encounters_grade	870	1688	3315
social_distancing_travel_distance_grade	860	1682	3310
daily_state_test	905	1791	3572
precipitation	1727	3456	6836
temperature	2368	4704	9330
ventilator_capacity_ratio	102	201	400
icu_beds_ratio	100	200	401
Religious_congregation_ratio	3	7	13
percent_insured	1	3	6
deaths_per_100000	143	262	543

**Table 4 entropy-24-00533-t004:** Results obtained for varying size test dataset.

Test Dataset Size	Imputation Method	Mean Absolute Error			Mean Squared Error		
		XGB	SVR	RFR	XGB	SVR	RFR
5000 Records	Proposed	60.81	202.01	112.8	8266.08	69,611.7	23,966
	Iterative	78.48	200.03	147.63	12,261.7	68,882.8	38,878.3
	KNN	82.3	201.91	147.15	12,972.8	69,768.5	37,811.4
	Simple Mean	79.78	197.48	146.88	12,197.3	68,160.8	37,889.9
	Dropping	68.08	197.37	145.84	8406.14	64,981.4	35,744.9
10,000 Records	Proposed	54.06	194.73	115.98	6046.26	63,853.1	23,256.3
	Iterative	72.84	196.45	145.58	10194	66,607.9	37,104.7
	KNN	75.58	198.2	148.12	11,154	67,537.5	38,554.6
	Simple Mean	73.36	192.69	146.96	10,372.3	65,122.9	38,134.9
	Dropping	68.08	197.37	146	8406.14	64,981.4	35,805.3
20,000 Records	Proposed	49.38	188.31	113.57	4473.7	59,422.4	23,298.4
	Iterative	72.69	192.51	145.98	9462.76	63,737.1	37,942.4
	KNN	75.01	193.38	145.21	9881.5	63,836.4	37,135.2
	Simple Mean	74.07	189.46	146.65	9695.8	62,288.6	37,528.1
	Dropping	68.08	197.37	146.02	8406.14	64,981.4	35,825.6

**Table 5 entropy-24-00533-t005:** Normalised results obtained for varying size test dataset.

Test Dataset Size	Imputation Method	Mean Absolute Error			Mean Squared Error		
		XGB	SVR	RFR	XGB	SVR	RFR
5000 Records	Iterative	0.775	1.010	0.764	0.674	1.011	0.616
	KNN	0.739	1	0.767	0.637	0.998	0.634
	Simple Mean	0.762	1.023	0.768	0.678	1.021	0.633
	Dropping	0.893	1.024	0.773	0.983	1.071	0.67
10,000 Records	Iterative	0.742	0.991	0.797	0.593	0.959	0.627
	KNN	0.715	0.982	0.783	0.542	0.945	0.603
	Simple Mean	0.737	1.011	0.789	0.583	0.981	0.610
	Dropping	0.794	0.987	0.794	0.719	0.983	0.650
20,000 Records	Iterative	0.679	0.978	0.778	0.473	0.932	0.614
	KNN	0.658	0.974	0.782	0.453	0.931	0.627
	Simple Mean	0.667	0.994	0.774	0.461	0.954	0.621
	Dropping	0.725	0.954	0.778	0.532	0.914	0.650

## Data Availability

Data used is referred from Haratian, A.; Fazelinia, H.; Maleki, Z.; Ramazi, P.; Wang, H.; Lewis, M.A.; Greiner, R.; Wishart, D. Dataset of COVID-19 outbreak and potential predictive features in the USA. *Data Brief* **2021**, *38*, 107360.

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
