# Peer review of "A Pragmatic Ensemble Strategy for Missing Values Imputation in Health Records"

_entropy, 2022, doi:10.3390/e24040533_

Round 1

Reviewer 1 Report

The authors proposed an ensemble approach for evaluating healthcare data which includes missing values. In particular, the authors have advanced the proposal to consider the contribution provided by three different data imputation approaches as a training set for regressor models. Among the "primitive" data imputation algorithms, the authors considered a KNN-based, an Iterative one, and one which considers the mean of values. The proposal has been evaluated by comparing the proposed ensemble approach with respect to the application of one imputation technique at a time for producing the training set, or by simply dropping the instances with a missing value for the purpose of building up the training set. Furthermore, the authors considered three different regression models, namely XGB Regressor, Support Vector Regressor, and the Random Forest regressor. With respect to the considered metrics, the evaluation has proved that the ensemble strategy is capable of outperforming the compared approaches, especially the ones where the primitive imputation technique has been applied. The paper is well written and certainly is focused on a current and important topic.

In order to improve the overall contribution derived by reading the article, however, I suggest the authors perform some minor adjustments.

-  Section 3 concerning the formalization of the ensemble approach is overly complex described, in particular, equations 4, 5, and 6 are provided but not adequately discussed, making it hard to understand the implication of such formulas on the proposed technique.

-  In Equation (2), the "score" function requires only one parameter as an input, i.e., b_i. However, in Figure 3 the same function is reported as requiring 2 input parameters. I suggest the authors be consistent with the notation in order to simplify the comprehension of the formalization.

-  Algorithm 1 is introduced but not properly described. The authors should perform an accurate description of the presented algorithm.

-  Table 2 should provide an overview of the total number of missing values for each test set.

-  Authors did not provide any discussion concerning the future works for their proposal. In particular, the authors should discuss if they consider their ensemble approach to be compatible with any data imputation technique and if they plan to evaluate such a scenario.

-  With respect to the previous point, the authors should include in their document other data imputation techniques existent in the literature, such as https://doi.org/10.1109/TKDE.2018.2883103, https://doi.org/10.1109/ICDE.2013.6544847, https://doi.org/10.5441/002/edbt.2022.05, and https://doi.org/10.1145/3394486.3403096

Author Response

Manuscript ID: entropy-1661434

Article Title: “A Pragmatic Ensemble Strategy for Missing Values Imputation in Health Records”

To: Ms. Max Ma, Assistant Editor (Entropy)

Re: Response to reviewers’ comments

Dear Ms. Max Ma,

We are delighted, Ms. Max Ma, that you offered us an invitation to revise our work for Entropy. Also, we truly appreciate you assigning such qualified reviewer to our manuscript. Their efforts and insights were a tremendous help to us during this revision. Again, we appreciate the opportunity to revise our work for consideration for publication in Entropy. We hope our revisions meet your approval. We are also grateful to the reviewer for the time and energy he/she expended on our behalf.

We are uploading (a) our point-by-point response to the comments (below) (response to reviewers), (b) an updated manuscript with track change mode indicating changes, and (c) a clean updated manuscript accepting all the tracked changes.

Best regards,

Authors.et.al.

Reviewer #1 Comments

Comment 1: Section 3 concerning the formalization of the ensemble approach is overly complex described, in particular, equations 4, 5, and 6 are provided but not adequately discussed, making it hard to understand the implication of such formulas on the proposed technique.

Answer 1:  As per the reviewer suggestion, content in the manuscript is revised and modified to improve the understanding of equations 4, 5, and 6

Modified text in the manuscript:

 is employed to prevent the challenge of overfitting by penalizing model intricacy as presented in equation (4) for an independent tree, t among the set of regression trees.

                                                        (4)

where and  are the regularization factors.  dictate if a particular node split depending on the anticipated loss minimization after the split, and  is L2 regularization on leaf weights.  and  are the number of leaves and scores on every leaf, respectively. The objective function can be approximated using a second-degree Taylor series [45]. Further, summation is a useful mechanism to train the ensemble model. Let  be an occurrence set of leaf j with the fixed structure . The equation (5) is used to calculate the optimum weights  of leaf j using first and second gradient orders of loss function and also the optimum value of associated loss function .

                             (5)

The first and second gradient orders of the loss function,  are  and  respectively. Further,  can be used to discover the quality score for t. The lower the score, the more accurate the model. Because computing all the tree topologies is impossible, a greedy approach is employed to tackle the issue, starting with only one leaf and repeatedly extending paths to the tree.

After splitting, let  and be the occurrence sets of the left and right nodes, respectively. If the original set is  = , the loss reduction following the split,  will be as presented in equation (6).

                             (6)

where the first term depicts the summation of score associated with left and right leaf, second term depicts the score associated with the original leaf i.e., leaf before the splitting operation is performed and  is the regularization term on additional leaf that will used further in the training process. In practice, this approach is commonly used to evaluate split candidates. During splitting, the XGB model employs many simple trees as well as the leaf node similarity score. 

Comment 2: In Equation (2), the "score" function requires only one parameter as an input, i.e., b_i. However, in Figure 3 the same function is reported as requiring 2 input parameters. I suggest the authors be consistent with the notation in order to simplify the comprehension of the formalization.

Answer 2:  To maintain the consistency in notation, we have made the desired changes in Figure 3 (Figure 4 in revised manuscript) and Figure 4 (Figure 5 in revised manuscript) as suggested.

Figures Modified in manuscript:

Figure 4.  Model Training Phase

Figure 5.  Imputation Phase

Comment 3: Algorithm 1 is introduced but not properly described. The authors should perform an accurate description of the presented algorithm.

Answer 3:  The detailed description of Algorithm 1 is included in the revised manuscript.

Text included in manuscript:

Algorithm #1 presents stepwise procedure of the proposed ensemble model. The algorithm has been partitioned into 3 sections i.e., variable declaration, generation of training dataset then training the model and applying trained model on the testing dataset.

  • In the first section (variable declaration), all the required dataset and matrices have been initialized.
  • In the second section, the algorithm performs two sequential tasks.
    1. The first task involves generation of training dataset using three imputation strategies i.e., Simple Imputation, kNN Imputation and Iterative Imputation, after applying imputation method on the training dataset, the resultant dataset is stored in , , and Now, for each attribute index present in , a corresponding matrix  is formed that comprises of four attributes (Simple, KNN, Iterative, and Actual). The first three attribute elements represented by vector B denoting the values of qth attribute’s elements imputed by Simple Imputation, KNN Imputation and Iterative Imputation Method and the fourth attribute element represented by vector A denoting the known value of qth attribute’s elements.
    2. The second task involves the training of a regressor model (XGB) using generated training dataset. The vector B and A are passed into XGBRegressor method for training the model and the trained resultant regressor associated with the qth attribute is represented by reg[q].
  • In the third section, the algorithm performs three sequential tasks.
    1. The first task involves the preprocessing of the testing dataset as done in previous section and transform testing dataset representation into matrix associated with each missing valued attribute (q).  matrix comprises of three attribute elements represented by vector  denoting the values of qth attribute’s elements imputed by Simple Imputation, KNN Imputation and Iterative Imputation Methods.
    2. The second task involves the prediction of missing values in testing dataset using trained regressor models (XGB) reg[q] associated with each missing valued attribute (q). The predicted values are stored in a vector .
    3. Lastly, the third task involves the substitution of imputed results of missing values associated with qth attribute as stored in into the actual dataset . After substitution the dataset is completed and no missing value is present in it

Comment 4: Table 2 should provide an overview of the total number of missing values for each test set.

Answer 4:  As recommended, Table 2 is enhanced with the details corresponding to frequency of missing values as well as non-missing values.

Text included in manuscript:

Table 1.  Instances holding one or more missing values in test dataset.

Test dataset size

Number of instances holding one or more missing values

Frequency of Missing Values

Frequency of Non-Missing Values

5000

3458

10123

279877

10000

6961

20045

559955

20000

13857

39722

1120278

Comment 5: Authors did not provide any discussion concerning the future works for their proposal. In particular, the authors should discuss if they consider their ensemble approach to be compatible with any data imputation technique and if they plan to evaluate such a scenario.

Answer 5:  A new section 5 “Discussion” is included in the manuscript that holds the details corresponding to the future work of the proposed model.

Text included in manuscript:

In current research authors are focused on establishing an ensemble technique for missing value imputation employing mean value, kNN and iterative imputation techniques. However, in near future authors aim to extend the current research on below-listed limiting parameters of the proposed model.

  • Functionally dependent domain: Current research is not exploiting the functional dependencies present in dataset for identification of missing values. The authors target to employ the devised ensemble strategy on other healthcare datasets including genomic-based and specific disease diagnosis based which may include the significance of attribute’s functional dependencies.
  • Intelligent selection of base predictors: The base predictors chosen in proposed model are fixed and thus, doesn’t consider other base predictors available. The authors intend to develop a system for intelligent selection and hybridization of the different base estimators on the basis of attributes domain dependency for instance, categorical data must be addressed by classification-based machine learning models and continuous data must be addressed by regression machine learning models. Further, the multiple stacking approach can be integrated for the meta learners in the proposed ensemble approach, wherein the XGB model can be replaced with the CNN based deep learning methods when handling complex healthcare datasets which can help in producing much better outcomes and can be more reliable in terms of performance.

Comment 6: With respect to the previous point, the authors should include in their document other data imputation techniques existent in the literature, such as

  • https://doi.org/10.1109/TKDE.2018.2883103,
  • https://doi.org/10.1109/ICDE.2013.6544847,
  • https://doi.org/10.5441/002/edbt.2022.05, and
  • https://doi.org/10.1145/3394486.3403096

Answer 6:  In current research authors are focused on establishing an ensemble technique for missing value imputation employing mean value, kNN and iterative imputation techniques. Author aims to employ strategies for dealing with function dependencies in near future (as specified in previous reviewer comment). Further, the literature suggested by reviewer is incorporated in the related work section of the revised manuscript.

Text included in manuscript:

For missing value imputation, the k nearest neighbours (kNN) approach is often employed. The kNN imputation method substitutes mean values from k closest neighbours for relevant attributes. Many studies have been conducted to increase kNN's imputation accuracy. To improve imputation efficiency, Song et al. took comparable neighbourhood into consideration [i]. More advanced single imputation strategies, including regression imputation and expectation-maximization (EM), can be used to resolve this issue [ii]. Regression models were used as a substitute to repair missing values in [iii]. Instead of attempting to deduce missing values, Song et. al. [iv] recommend first estimating lengths between absent and entire values, and then imputing values using inferred lengths. These techniques allocate a missing value by analysing the correlations in between the dependent attribute and the remaining parameters in the dataset. Chu et. al. [v] focused on data cleaning approaches including various functional dependencies in a unified framework. Breve et. al. [vi] proposed a novel data imputation technique based on relaxed functional dependencies that identifies value possibilities that effectively ensure data integrity. However, in the case of healthcare data, we often encounter temporal functional dependencies for the data of patient collected for a time span [vii].

References:

  1. Song, S.; Sun, Y.; Zhang, A.; Chen, L.; Wang, J. Enriching data imputation under similarity rule constraints. IEEE transactions on knowledge and data engineering2018, 32(2), 275-287.
  2. Hu, Z.; Melton, G.B.; Arsoniadis, E.G.; Wang, Y.; Kwaan, M.R.; Simon, G.J. Strategies for handling missing clinical data for automated surgical site infection detection from the electronic health record. Journal of biomedical informatics, 2017, 68, pp. 112-120.
  • Nikfalazar, S.; Yeh, C.H.; Bedingfield, S.; Khorshidi, H.A. A new iterative fuzzy clustering algorithm for multiple imputation of missing data. In 2017 IEEE International Conference on Fuzzy Systems (FUZZ-IEEE), 2017, pp. 1-6. IEEE.
  1. Song, S.; Sun, Y. Imputing various incomplete attributes via distance likelihood maximization. In Proceedings of the 26th ACM SIGKDD International Conference on Knowledge Discovery & Data Mining, 2020, pp. 535-545.
  2. Chu, X.; Ilyas, I. F.; Papotti, P. Holistic data cleaning: Putting violations into context. In 2013 IEEE 29th International Conference on Data Engineering (ICDE), 2013, pp. 458-469. IEEE.
  3. Breve, B.; Caruccio, L.; Deufemia, V.; Polese, G. RENUVER: A Missing Value Imputation Algorithm based on Relaxed Functional Dependencies, open Proceedings, 2022.
  • Combi, C.; Mantovani, M.; Sabaini, A.; Sala, P.; Amaddeo, F.; Moretti, U.; Pozzi, G. Mining approximate temporal functional dependencies with pure temporal grouping in clinical databases. Computers in biology and medicine2015, 62, 306-324.

Reviewer 2 Report

very interesting and detailed investigation

important for digitalization in healthcare

Author Response

Manuscript ID: entropy-1661434

Article Title: “A Pragmatic Ensemble Strategy for Missing Values Imputation in Health Records”

To: Ms. Max Ma, Assistant Editor (Entropy)

Re: Response to reviewers’ comments

Dear Ms. Max Ma,

We are delighted, Ms. Max Ma, that you offered us an invitation to revise our work for Entropy. Also, we truly appreciate you assigning such qualified reviewer to our manuscript. Their efforts and insights were a tremendous help to us during this revision. Again, we appreciate the opportunity to revise our work for consideration for publication in Entropy. We hope our revisions meet your approval. We are also grateful to the reviewer for the time and energy he/she expended on our behalf.

We are uploading (a) our point-by-point response to the comments (below) (response to reviewers), (b) an updated manuscript with track change mode indicating changes, and (c) a clean updated manuscript accepting all the tracked changes.

Best regards,

Authors.et.al.

Reviewer #2 Comments

Comment 1: very interesting and detailed investigation, important for digitalization in healthcare

Answer 1:  we are thankful to the reviewer for the appreciation.

Reviewer 3 Report

This study evaluates different imputation and regression procedures identified based on regressor performance and computational expense to fix the issues of missing values in both training and testing datasets. In the context of healthcare, several procedures are introduced for dealing with missing values. According to the study, there is still a discussion concerning which imputation strategies are better in specific cases. This research proposes an ensemble imputation model that is educated to use a combination of simple mean imputation, k-nearest neighbour imputation, and iterative imputation methods and then leverages them in a manner that the ideal imputation strategy is opted among them based on attribute correlations on missing value features. The performance metrics have been generated using the eXtreme gradient boosting regressor, random forest regressor, and support vector regressor. The findings demonstrate that the proposed ensemble imputation model outperforms listed imputation approaches as well as the approach of dropping records holding missing values in terms of accuracy. 

The paper discusses a hot topic of the related literature that the reader of this Journal would like to read.

While this is a very interesting paper, I think it is necessary to address some concerns before publication.

Some improvements should be done for a better comprehensive reading.
I would suggest to the authors that include some discussion about explainability for the results. Also, the following issues should be improved:

  1. In the abstract, the novelty of this research should be discussed.
  2. In the introduction, the motivation and contribution of this paper should be given.
  3. Since the literature review is quite poor and to support several assertions about ensemble learning, the authors are adivsed to use the following references:
    • Troussas, C.; Krouska, A.; Sgouropoulou, C.; Voyiatzis, I. Ensemble Learning Using Fuzzy Weights to Improve Learning Style Identification for Adapted Instructional Routines. Entropy 202022, 735. https://doi.org/10.3390/e22070735

    • Zhao, D.; Wang, X.; Mu, Y.; Wang, L. Experimental Study and Comparison of Imbalance Ensemble Classifiers with Dynamic Selection Strategy. Entropy 202123, 822. https://doi.org/10.3390/e23070822

    • Rahimi N, Eassa F, Elrefaei L. One- and Two-Phase Software Requirement Classification Using Ensemble Deep Learning. Entropy. 2021; 23(10):1264. https://doi.org/10.3390/e23101264

  4. In Section 3, a better description of the the research methodology (maybe accompanied with a corresponding schema) could be helpful.
  5. Some more information about the Regressor/ Imputation Methods  could be included.
  6. The conclusions should lead to new knowledge. Also, limitations of this research are missing at the moment.

Concluding, the structure of paper is good, but the main contributions of the paper do not add significant value to the existing body of knowledge in the related subject area.
A suggested contribution is to have a discussion section to compare the presented work with the related work in the literature.

Overall, the paper is well organized. However, it lacks critical discussion in contrast with the related work in the literature and does not provide major contributions to the field.

Author Response

Manuscript ID: entropy-1661434

Article Title: “A Pragmatic Ensemble Strategy for Missing Values Imputation in Health Records”

To: Ms. Max Ma, Assistant Editor (Entropy)

Re: Response to reviewers’ comments

Dear Ms. Max Ma,

We are delighted, Ms. Max Ma, that you offered us an invitation to revise our work for Entropy. Also, we truly appreciate you assigning such qualified reviewer to our manuscript. Their efforts and insights were a tremendous help to us during this revision. Again, we appreciate the opportunity to revise our work for consideration for publication in Entropy. We hope our revisions meet your approval. We are also grateful to the reviewer for the time and energy he/she expended on our behalf.

We are uploading (a) our point-by-point response to the comments (below) (response to reviewers), (b) an updated manuscript with track change mode indicating changes, and (c) a clean updated manuscript accepting all the tracked changes.

Best regards,

Authors.et.al.

Reviewer #3 Comments

Comment 1: I would suggest to the authors that include some discussion about explainability for the results.

Answer 1:  A new section 5 “Discussion” is included in the manuscript for better explanability of results as recommended by reviewer.

Text included in manuscript:

  1. Discussion

After analyzing the evaluation metrics generated by three regressor models, it has been found that the proposed ensemble strategy is the most suitable option for the imputation of missing values. The Imputed dataset produced by the Proposed Ensemble approach when passed to XGB Regressor for performance evaluation results in the least mean absolute error i.e., 60.81, 54.06, and 49.38 and least mean squared error i.e., 8266.08, 6046.26, and 4473.7 in all the three test cases considered. Similarly, when the same dataset is passed to the RFR model, the model gives the least mean absolute error i.e., 112.8, 115.98 and 113.57and the least mean squared error i.e., 23966, 23256.3 and 23298.47 in all the three test cases considered. However, when the same imputed dataset is passed to SVR Model then in one of the test cases i.e., with 20000 records, it gives the least mean absolute error i.e., 188.31 and in two cases i.e., with 10000 and 20000 records, it gives least means squared error i.e., 63853.1 and 59422.4 respectively as represented in Figure 8.

For the comparison of state-of-the-art missing value handling strategies such as simple imputation, KNN imputation, iterative imputation and dropping the missing value contained instances method, normalized error results have been calculated using equation (11) and equation (12) with respect to the proposed imputation method as depicted in Table 5. It has been observed that the approach of dropping the instances with missing values is the closest missing value handling method to the proposed ensemble model as it results in the normalized error estimate in the range of 0.7 and 1.0 in all three considered test cases. But the method reduces the dataset size, thus it should not be preferred for large and crucial datasets.

On the other hand, among simple mean, KNN and iterative methods, Iterative Imputation is closest to the proposed imputation method having a normalized MAE of 0.775, 0.742, and 0.679 in the three considered test cases i.e., 5000,10000, and 20000 records respectively and a normalized MSE of 0.593 and 0.473 in two test cases i.e., 10000 and 20000 records respectively as computed by XGB Regressor Model. On the contrary, the Simple Mean Imputation method is closest to the proposed imputation method having a normalized MAE 1.023, 1.011, and 0.994 and a normalized MSE 1.021, 0.981, and 0.954 in the three considered test cases i.e., 5000,10000, and 20000 records, respectively as predicted by SVR Model and a normalized MAE and normalized MSE of 0.768 and 0.678 as predicted by RFR and XGB Model. Similarly, the KNN Imputation method closest to the proposed imputation method having a normalized MAE 0.782 in one test case i.e., 20000 records and a normalized MSE of 0.634 and 0.627 in the two considered test cases i.e., 5000 and 20000 records respectively as predicted by RFR Model. Hence it can be said, when the dataset size is small and has fewer missing values, dropping the records holding missing values seems most suitable approach, as predicted by almost all three regression models and with a large dataset size Simple Mean, KNN and Iterative Method gives equivalent result in most of the cases but could not match with the performance of the proposed ensemble strategy as estimated by considered regressor models.

In current research authors are focused on establishing an ensemble technique for missing value imputation employing mean value, kNN and iterative imputation techniques. However, in near future authors aim to extend the current research on below-listed limiting parameters of the proposed model.

  • Functionally dependent domain: Current research is not exploiting the functional dependencies present in dataset for identification of missing values. The authors target to employ the devised ensemble strategy on other healthcare datasets including genomic-based and specific disease diagnosis based which may include the significance of attribute’s functional dependencies.
  • Intelligent selection of base predictors: The base predictors chosen in proposed model are fixed and thus, doesn’t consider other base predictors available. The authors intend to develop a system for intelligent selection and hybridization of the different base estimators on the basis of attributes domain dependency for instance, categorical data must be addressed by classification-based machine learning models and continuous data must be addressed by regression machine learning models. Further, the multiple stacking approach can be integrated for the meta learners in the proposed ensemble approach, wherein the XGB model can be replaced with the CNN based deep learning methods when handling complex healthcare datasets which can help in producing much better outcomes and can be more reliable in terms of performance.

Comment 2:  In the abstract, the novelty of this research should be discussed.

Answer 2:  As per the suggestion of reviewer, we have enriched the abstract with following content to address the novelty of the current research.

Text included in manuscript: We introduce a unique Ensemble Strategy for Missing Value to analyse healthcare data with considerable missing values to identify unbiased and accurate prediction statistical modelling. The current study uses real-world healthcare data to conduct experiments and simulations of data with varying feature-wise missing frequencies indicating that the proposed technique surpasses standard missing value imputation approaches as well as the approach of dropping records holding missing values in terms of accuracy.

Comment 3: In the introduction, the motivation and contribution of this paper should be given.

Answer 3:  The manuscript is enhanced with a new sub-section 1.1 “Motivation” to highlight the motivation of current research. In addition, contributions of current research are re-organized for better understanding in sub-section 1.5 “Research Contributions”

Text included in manuscript:

1.1. Motivation

In healthcare prediction, missing data raises serious analytical difficulties. If missing data isn't treated seriously, it might lead to skewed forecasts. The challenge of dealing with missing values in massive medical databases still need more efforts to be addressed [a]. To minimise the harm to data processing outcomes, it is advisable to integrate multiple known ways for addressing missing data (or design new ones) for each system. The demand for missing data imputation approaches that result in improved imputed values than conventional systems with greater precision and smaller biases is the driving force behind this study.

References:

  1. Mirkes, E.M.; Coats, T.J.; Levesley, J.; Gorban, A. N. Handling missing data in large healthcare dataset: A case study of unknown trauma outcomes. Computers in biology and medicine2016, 75, 203-216.

Text reorganized in manuscript:

1.5. Research Contributions

Current research provides following key research contributions.

  1. We introduce a unique Ensemble Strategy for Missing Value to analyse healthcare data with considerable missing values to identify unbiased and accurate prediction statistical modelling. Overall, there are four computational benefits of the suggested model:
  • It can analyse huge amounts of health data with substantial missing values and impute them more correctly than standalone imputation procedures such as the k-nearest neighbour approach, iterative method, and so on.
  • It can discover essential characteristics in a dataset with many missing values.
  • It tackles the performance glitches of developing a single predictor to impute missing values, such as high variance, feature bias, and lack of precision.
  • Fundamentally, it employs an extreme gradient boosting method, which includes L1 (Lasso Regression) and L2 (Ridge Regression) regularization to avoid overfitting.
  1. The current study uses real-world healthcare data (snapshot presented in Error! Reference source not found.) to conduct experiments and simulations of data with varying feature-wise missing frequencies indicating that the proposed technique surpasses standard missing value imputation approaches.

Comment 4: Since the literature review is quite poor and to support several assertions about ensemble learning, the authors are advised to use the following references:

  • Troussas, C.; Krouska, A.; Sgouropoulou, C.; Voyiatzis, I. Ensemble Learning Using Fuzzy Weights to Improve Learning Style Identification for Adapted Instructional Routines. Entropy202022, 735. https://doi.org/10.3390/e22070735
  • Zhao, D.; Wang, X.; Mu, Y.; Wang, L. Experimental Study and Comparison of Imbalance Ensemble Classifiers with Dynamic Selection Strategy. Entropy202123, 822. https://doi.org/10.3390/e23070822
  • Rahimi N, Eassa F, Elrefaei L. One- and Two-Phase Software Requirement Classification Using Ensemble Deep Learning. Entropy. 2021; 23(10):1264. https://doi.org/10.3390/e23101264

Answer 4:  The suggested literature is incorporated in the related work section of the revised manuscript.

Text included in manuscript:

Ensemble approaches have been utilized in a variety of domain to improve the accuracy of system. Troussas et al. [b] suggested an ensemble classification approach that uses the support vector machine, naive bayes, and KNN classifiers in combination with a majority voting mechanism to categorise learners into appropriate learning styles. The model is first trained using a collection of data, and then the category of the occurrence is forecasted using the base classifiers with the majority of votes. Zaho et al. [c] devised an ensemble technique by integrating patch learning with dynamic selection ensemble classification, wherein the miscategorised data have been used to educate patch models in order to increase the variety of base classifiers. Rahimi et al. [d] used ensemble deep learning approaches to construct a classification model that improved the accuracy and reliability for classifying software requirement specifications.

References:

  1. Troussas, C.; Krouska, A.; Sgouropoulou, C.; Voyiatzis, I. Ensemble Learning Using Fuzzy Weights to Improve Learning Style Identification for Adapted Instructional Routines. Entropy202022, 735. https://doi.org/10.3390/e22070735
  2. Zhao, D.; Wang, X.; Mu, Y.; Wang, L. Experimental Study and Comparison of Imbalance Ensemble Classifiers with Dynamic Selection Strategy. Entropy202123, 822. https://doi.org/10.3390/e23070822
  3. Rahimi N, Eassa F, Elrefaei L. One- and Two-Phase Software Requirement Classification Using Ensemble Deep Learning. Entropy. 2021; 23(10):1264. https://doi.org/10.3390/e23101264

Comment 5: In Section 3, a better description of the research methodology (may be accompanied with a corresponding schema) could be helpful.

Answer 5:  Content in Section 3 is enhanced with more details of the research methodology and a new Figure 2 is added to provide more clarity.

Text included in manuscript: Ensemble learning is an amalgamation of various machine learning technique that takes contemplates the estimate of various base machine learning models (base estimators) in order to achieve better predictive performance. As base estimator, one can implement any machine learning algorithms. If the nature of considered base learners are homogeneous then the ensemble strategy is termed as a homogeneous ensemble learning method otherwise the ensemble strategy is termed as non-homogeneous or heterogeneous. The Ensemble Machine learning can be constructed on three sorts of mechanisms viz. bootstrap aggregation (Bagging), boosting, and stacking. Bootstrap aggregation comprises of independently learning weak learners (base estimators) outcome is the average of resultants calculated by different weak learning. While in boosting mechanism, the base estimators are summarized one after the another then resultant is generated as the weighted average of base estimators’ outcomes. On the other Hand, stacking ensemble mechanism fed the same data to all chosen base estimators then trains an additional machine learning model called as meta-learner to upgrade model’s overall performance. In this research, the authors have employed the stacking mechanism of ensemble strategy in order to devise a novel methodology of missing data imputation for Health Informatics. This research will be using different stand-alone imputation as individual base estimators in level 1 and then combines the outcomes of these base estimators and fed them to a meta learner machine learning model in Level 2 to make the final predictions. The Figure 2 illustrates the conceptual schema of the proposed ensemble strategy.

Figure 2. Conceptual Schema of Proposed Ensemble Approach based on Stacking Mechanism.

Comment 6: Some more information about the Regressor/ Imputation Methods could be included.

Answer 6:  To address the reviewer concern, a new sub-section 4.2 “Regressor Models” is incorporated in the revised manuscript.

Text included in manuscript:

4.2 Regressor Models

For determining the performance of the proposed ensemble framework, the authors have selected three regression models i.e., Support Vector Regressor (SVR), Random Forest Regressor (RFR), and eXtreme Gradient Boost Regressor (XGBR). These Regression Models are built to check the performance of different missing data handling methodologies discussed in the paper (i.e., Proposed Ensemble Imputation Method, Simple Mean Imputation Method, kNN Imputation Method and Iterative Imputation Method). The covid_19_deaths attribute is chosen as the target attribute to train and test these models because it has no missing values and it also happens to be the target variable in the dataset [e]. The Regressor models are briefly illustrated as follows:

  1. eXtreme Gradient Boost Regressor (XGBR): XGBoost is a tree-based enactments of gradient boosting machines (GBM) utilized for supervised machine learning. XGBoost is a widely used Machine Learning algorithm in Kaggle Competitions [f] and is favored by data scientists as its high execution speed beats principal computations [g]. The key concept behind boosting regression strategy is the consecutive construction of subsequent trees from a rooted tree such that each successive tree diminishes the errors of the tree previous to it, so that the newly formed subsequent trees will update the preceding residuals to decrease the cost function error. In this research the XGB Regressor Model has a maximum tree depth of 10, L1 and L2 regularization term on weights are set as default i.e., 0 and 1 respectively.
  2. Random Forest Regressor (RFR): Random Forest is an ensemble tree-based regression methodology proposed by Leo Breiman, it is a substantial alteration of bootstrap aggregating that builds a huge assemblage of contrasting trees, after that aggregates them [h]. A random forest predictor comprises an assemblage of unpremeditated regression trees as base , where ,, are independent and identically distributed (IID) outcomes of a randomizing variable and . An aggregated regression estimate is evaluated by combining all these random trees by using formula , where  denotes expectation w.r.t. the random variable conditionally on A and the dataset . In this research the maximum depth of RFR tree is tuned as 5 and other parameters such minimum sample split and number of trees are kept as default i.e., 2 and 1000 respectively.
  3. Support Vector Regressor (SVR): Support Vector Machine (SVM) used for regression analysis is named as support vector regressor (SVR) [i]. In SVR, the input values are mapped into a higher-dimensional space by some non-linear functions called as kernel functions [j,k] so as to make the model linearly separable for making predictions. The SVR model is trained by a structural risk minimization (SRM) principle [l] to perform regression. This minimizes the VC dimension [m] as a replacement for minimizing the mean absolute value of error or the squared error. In this research, SVR uses the radial basis function as kernel and a regularization parameter (C) of 1.5.

References:

  1. Haratian, A.; Fazelinia, H.; Maleki, Z.; Ramazi, P.; Wang, H.; Lewis, M.A.; Greiner, R.; Wishart, D. Dataset of COVID-19 outbreak and potential predictive features in the USA,” Data in Brief, 2021, 38.
  2. Chen, M.; Liu, Q.; Chen, S.; Liu, Y.; Zhang, C.H.; Liu, R. XGBoost-based algorithm interpretation and application on post-fault transient stability status prediction of power system. IEEE Access2019, 7, 13149-13158.
  3. Chen T.; Guestrin, C. Xgboost: A scalable tree boosting system. In Proceedings of the 22nd acm sigkdd international conference on knowledge discovery and data mining, New York, NY, USA, 13 August 2016.
  4. Breiman, L. Random forests. Machine learning2001, 45(1), 5-32.
  5. Drucker, H.; Burges, C.J.; Kaufman, L.; Smola, A.; Vapnik, V. Support vector regression machines. Advances in neural information processing systems1996, 9.
  6. Wu, M.C.; Lin, G.F.; Lin, H.-Y. Improving the forecasts of extreme streamflow by support vector regression with the data extracted by self-organizing map. Process. 2014, 28, 386–397.
  7. Wu, C.L.; Chau, K.W.; Li, Y.S. River stage prediction based on a distributed support vector regression. Hydrol. 2008, 358, 96–111.
  8. Yu, P.S.; Chen, S.T.; Chang, I.F. Support Vector Regression for Real-Time Flood Stage Forecasting. Hydrol. 2006, 328, 704–716.
  9. Viswanathan, M.; Kotagiri, R. Comparing the performance of support vector machines to regression with structural risk minimisation. In International Conference on Intelligent Sensing and Information Processing, 2004. doi: https://doi.org/1109/ICISIP.2004.1287698

Comment 7: The conclusions should lead to new knowledge. Also, limitations of this research are missing at the moment.

Answer 7:  A new section 5 “Discussion” is included that holds the details about limitations of current research.

Text included in manuscript:

In current research authors are focused on establishing an ensemble technique for missing value imputation employing mean value, kNN and iterative imputation techniques. However, in near future authors aim to extend the current research on below-listed limiting parameters of the proposed model.

  • Functionally dependent domain: Current research is not exploiting the functional dependencies present in dataset for identification of missing values. The authors target to employ the devised ensemble strategy on other healthcare datasets including genomic-based and specific disease diagnosis based which may include the significance of attribute’s functional dependencies.
  • Intelligent selection of base predictors: The base predictors chosen in proposed model are fixed and thus, doesn’t consider other base predictors available. The authors intend to develop a system for intelligent selection and hybridization of the different base estimators on the basis of attributes domain dependency for instance, categorical data must be addressed by classification-based machine learning models and continuous data must be addressed by regression machine learning models. Further, the multiple stacking approach can be integrated for the meta learners in the proposed ensemble approach, wherein the XGB model can be replaced with the CNN based deep learning methods when handling complex healthcare datasets which can help in producing much better outcomes and can be more reliable in terms of performance.

Comment 8: Concluding, the structure of paper is good, but the main contributions of the paper do not add significant value to the existing body of knowledge in the related subject area.
A suggested contribution is to have a discussion section to compare the presented work with the related work in the literature.

Answer 8:  A new section 5 “Discussion” is included in the manuscript as recommended by the reviewer.

Text included in manuscript:

  1. Discussion

After analyzing the evaluation metrics generated by three regressor models, it has been found that the proposed ensemble strategy is the most suitable option for the imputation of missing values. The Imputed dataset produced by the Proposed Ensemble approach when passed to XGB Regressor for performance evaluation results in the least mean absolute error i.e., 60.81, 54.06, and 49.38 and least mean squared error i.e., 8266.08, 6046.26, and 4473.7 in all the three test cases considered. Similarly, when the same dataset is passed to the RFR model, the model gives the least mean absolute error i.e., 112.8, 115.98 and 113.57and the least mean squared error i.e., 23966, 23256.3 and 23298.47 in all the three test cases considered. However, when the same imputed dataset is passed to SVR Model then in one of the test cases i.e., with 20000 records, it gives the least mean absolute error i.e., 188.31 and in two cases i.e., with 10000 and 20000 records, it gives least means squared error i.e., 63853.1 and 59422.4 respectively as represented in Figure 8.

For the comparison of state-of-the-art missing value handling strategies such as simple imputation, KNN imputation, iterative imputation and dropping the missing value contained instances method, normalized error results have been calculated using equation (11) and equation (12) with respect to the proposed imputation method as depicted in Table 5. It has been observed that the approach of dropping the instances with missing values is the closest missing value handling method to the proposed ensemble model as it results in the normalized error estimate in the range of 0.7 and 1.0 in all three considered test cases. But the method reduces the dataset size, thus it should not be preferred for large and crucial datasets.

On the other hand, among simple mean, KNN and iterative methods, Iterative Imputation is closest to the proposed imputation method having a normalized MAE of 0.775, 0.742, and 0.679 in the three considered test cases i.e., 5000,10000, and 20000 records respectively and a normalized MSE of 0.593 and 0.473 in two test cases i.e., 10000 and 20000 records respectively as computed by XGB Regressor Model. On the contrary, the Simple Mean Imputation method is closest to the proposed imputation method having a normalized MAE 1.023, 1.011, and 0.994 and a normalized MSE 1.021, 0.981, and 0.954 in the three considered test cases i.e., 5000,10000, and 20000 records, respectively as predicted by SVR Model and a normalized MAE and normalized MSE of 0.768 and 0.678 as predicted by RFR and XGB Model. Similarly, the KNN Imputation method closest to the proposed imputation method having a normalized MAE 0.782 in one test case i.e., 20000 records and a normalized MSE of 0.634 and 0.627 in the two considered test cases i.e., 5000 and 20000 records respectively as predicted by RFR Model. Hence it can be said, when the dataset size is small and has fewer missing values, dropping the records holding missing values seems most suitable approach, as predicted by almost all three regression models and with a large dataset size Simple Mean, KNN and Iterative Method gives equivalent result in most of the cases but could not match with the performance of the proposed ensemble strategy as estimated by considered regressor models.

In current research authors are focused on establishing an ensemble technique for missing value imputation employing mean value, kNN and iterative imputation techniques. However, in near future authors aim to extend the current research on below-listed limiting parameters of the proposed model.

  • Functionally dependent domain: Current research is not exploiting the functional dependencies present in dataset for identification of missing values. The authors target to employ the devised ensemble strategy on other healthcare datasets including genomic-based and specific disease diagnosis based which may include the significance of attribute’s functional dependencies.
  • Intelligent selection of base predictors: The base predictors chosen in proposed model are fixed and thus, doesn’t consider other base predictors available. The authors intend to develop a system for intelligent selection and hybridization of the different base estimators on the basis of attributes domain dependency for instance, categorical data must be addressed by classification-based machine learning models and continuous data must be addressed by regression machine learning models. Further, the multiple stacking approach can be integrated for the meta learners in the proposed ensemble approach, wherein the XGB model can be replaced with the CNN based deep learning methods when handling complex healthcare datasets which can help in producing much better outcomes and can be more reliable in terms of performance.

******End of the document******

Round 2

Reviewer 1 Report

The authors performed all the suggested improvements to the paper. The document can be accepted in its current form.

Reviewer 3 Report

The paper has been improved significantly and can be accepted for publication in its surrent form.